# You Don't Have to Be the One Doing Evil! Locate a Scapegoat within Multi-Agent Systems for Executing Covert Attacks

## Abstract

While LLM-based Multi-Agent Systems (MAS) excel at complex tasks through collaboration, their interactive process introduces security vulnerabilities. Existing research has shown that malicious agents can disrupt task execution by injecting prompts or manipulating the shared memory pool. However, these methods often require the malicious agent to perform direct and traceable actions, making them susceptible to detection by auditors through log reviews. Unlike prior approaches, this paper explores a more covert attack strategy, namely the *Scapegoating* attack, where the malicious agent induces downstream agents in the MAS to output content that sabotages task completion, rather than directly performing the sabotage itself. To conduct such a covert attack, we further design SGoatMAS, a systemic chain poisoning framework. The core idea is to view the entire multi-agent workflow as an integrated "System-level Chain of Thought" and propose a new metric, Vulnerability-to-Risk ratio, to identify both the most vulnerable link and the most plausible scapegoat agent in the chain. Then, we propose conducting attacks on the selected link and agent. Extensive experiments demonstrate that SGoatMAS not only achieves significant attack performance but also maintains exceptionally low detection rates. This work points to a new direction for future research into the security and defense of Multi-Agent Systems.

## 1 Introduction

Recently, Multi-Agent collaborative Systems (MAS) (Wang et al., 2024a; Guo et al., 2024), where multiple Large Language Models (LLMs) agents collaborate to achieve shared goals, have garnered significant attention due to their remarkable proficiency in a wide array of tasks such as scientific discovery and code generation (Brown et al., 2020). MAS typically allows different agents to organize in some specific structure and work in a given workflow (Wu et al.; Hong et al., 2024; Qian et al., 2023; Park et al., 2023) when carrying out some plans to achieve the given target.

However, MAS not only changes the way tasks are processed but also affects their robustness against adversarial manipulations. In particular, one malicious agent may launch direct prompt injection attacks against another within a shared conversational space (Debenedetti et al., 2024), or may poison a shared resource such as the shared memory pool (Chen et al., 2024). While effectively prohibiting the achievement of the target, existing methods often require the malicious agent to conduct direct and traceable actions, leaving a clear log. This behavior significantly increases the risk of the malicious agent being exposed, especially after human auditing. Therefore, the question this paper aims to address is: *Can a malicious agent carry out covert attacks—achieving successful sabotage of the task objective while remaining undetected and avoiding exposure?*

Our answer to the above question is affirmative. Our new insight is the *Scapegoating* attack, a more dangerous class of attack than traditional direct attacking methods. More specifically, we propose that the malicious agent should induce downstream agents in the MAS to output content that sabotages task completion, rather than directly performing the sabotage itself. We refer to such induced agents as *scapegoats*, which are more likely to be identified by auditors as the primary culprits during subsequent log reviews or audits, thereby allowing the true attacker to remain overlooked.

Compared to traditional MAS attacks, this approach is not only more covert but also prone to "collateral damage" to benign agents, potentially causing more severe disruption.

To implement such a Scapegoating attack to MAS, we propose SGoatMAS, a systemic chain poisoning framework. Specifically, we treat the entire multi-agent workflow as a distributed, System-level Chain of Thought, enabling us to pinpoint systemic logical vulnerabilities. To identify the optimal point in the MAS workflow to inject a logical fallacy, we propose the Vulnerability-to-Risk Ratio (VRR) to measure the vulnerability of the system. Then, we leverage the reasoning capabilities elicited by Chain-of-Thought (CoT) prompting (Wei et al., 2022) to generate these logic-based payloads. Our main contributions are as follows:

- We propose the concept of Scapegoating attack to MAS. As is known to us, this is the first covert attack on MAS, potentially causing more severe disruption, which should raise sufficient attention to the design of defense strategies.
- We design SGoatMAS, an implementation of the Scapegoating attack, leveraging the idea of the Chain of Thought in a systematic level by dissecting reasoning links of the whole MAS workflow.
- We conduct a comprehensive experimental validation of SGoatMAS on a suite of challenging benchmarks (MMLU (Hendrycks et al., 2020), MMLU-Pro (Wang et al., 2024d), GSM8K (Cobbe et al., 2021), Arithmetic (Du et al., 2023), HumanEval (Chen et al., 2021)). Our experiments quantify the attack's effectiveness and stealth across diverse workflow topologies, providing a crucial red-teaming analysis of the systemic risks of implicit trust in the next generation of AI.

## 2 RELATED WORK

### 2.1 LLM-BASED MULTI-AGENT SYSTEMS

The paradigm of multi-agent systems (Wooldridge, 2009) has been revitalized by the capabilities of LLMs. A significant body of recent work has focused on creating architectural frameworks for collaboration. Systems like MetaGPT (Hong et al., 2024) and ChatDev (Qian et al., 2023) impose structured, role-based workflows to mimic human organizations and enhance efficiency. In contrast, frameworks like AutoGen (Wu et al.) and AgentVerse (Chen et al., 2023) provide more flexible, conversational platforms where agents can dynamically collaborate. Research has also explored the emergent behaviors that arise from these interactions, from believable social simulations (Park et al., 2023) to complex problem-solving patterns involving self-reflection and criticism (Huang et al., 2022; Yao et al., 2023). While this foundational work establishes the performance capabilities of cooperative MAS, the security vulnerabilities inherent in their trusted communication channels remain largely unexplored. Our work pivots from performance analysis to an offensive security perspective, investigating how these collaborative structures can be subverted from within.

### 2.2 ATTACK METHODOLOGIES IN LLM-INTEGRATED SYSTEMS

Research into LLM security has identified several key threat vectors. Prompt-based Attacks, including direct and indirect prompt injection, remain a primary concern, allowing attackers to hijack an agent's control flow or bypass safety measures (Pedro et al., 2023; Shi et al., 2024; Greshake et al., 2023). The creation of such malicious prompts can even be automated through optimization techniques (Zou et al., 2023; Shi et al., 2024). Another major threat is Data Poisoning, which compromises the model itself during training or fine-tuning to create hidden backdoors (Wang et al., 2024c; Zhang et al., 2024). A nascent but growing area of research is focused on attacks specifically within MAS. Initial studies have demonstrated direct, inter-agent prompt injections in conversational settings (Lee & Tiwari, 2024) and the poisoning of shared resources like system memory or tools (Chen et al., 2024). Other work has shown that agents are susceptible to persuasion and deception (Liu et al., 2025; Wang et al., 2024b; Hagendorff, 2024). Our work is distinct from these existing approaches. While prior MAS attacks are often direct and risk creating detectable anomalies in communication logs, our SGoatMAS framework is designed for stealth. It focuses on inducing errors through seemingly benign, role-consistent information, thereby evading anomaly detection and framing an innocent agent.

## 3 PRELIMINARIES AND PROBLEM FORMULATION

To formally reason about inducement and scapegoating attacks, we must first establish a precise model of the MAS workflow and the capabilities of the internal adversary. Our formulation is designed to be general enough to capture a wide variety of collaborative LLM architectures while being specific enough to ground our attack framework. We begin by modeling the MAS workflow as a directed acyclic graph (DAG), a standard representation for systems with defined stages and information flow.

**Definition 1 (Multi-Agent System Workflow).** A MAS workflow is a tuple $G = (A, E, \tau)$, where $A = \{a_1, a_2, \ldots, a_n\}$ is a finite set of $n$ LLM-based agents. The information flow protocol between these agents is defined by a set of directed edges, $E \subseteq A \times A$. The existence of an edge

$$e = (a_i, a_j) \in E \tag{1}$$

satisfies that the output of agent $a_i$ serves as a primary input for agent $a_j$. Each agent $a_i \in A$ is assigned a specific function within the system, its role, which is defined by the mapping $\tau : A \to \Sigma$, where $\Sigma$ is the set of all possible roles (e.g., $\Sigma = \{\text{Planner, Coder, Verifier}\}$). The output of an agent $a_i$ at a given timestep $t$ is denoted by $c_i^{(t)}$.

Within this formally defined system operates a malicious agent, $a_m \in A$. Our attacker model assumes that this agent has already compromised a position within the trusted set of agents $A$. This initial compromise could be the result of a user unknowingly integrating a malicious third-party service (following the MCP pattern) or a supply chain attack on one of the system's legitimate agent components. The malicious agent's behavior is characterized by its strategic and covert nature.

**Definition 2 (Attacker Model).** The internal malicious agent, $a_m \in A$, is defined by the tuple $(\mathcal{O}_m, S)$, where $\mathcal{O}_m$ is the malicious objective and $S$ is the attack strategy. The objective $\mathcal{O}_m$ is to manipulate the final system output, $c_{\text{final}}$, such that a specific, malicious condition is met. The strategy $S$ is based on stealth and inducement, mandating that the agent's output, $c_m$, must be se-mantically consistent with its assigned role, $\tau(a_m)$ (Role-Consistency). The payload $c_m$ is not necessarily factually incorrect but is crafted to exploit a cognitive or logical vulnerability in a down-stream agent (Inducement). Critically, the strategy aims for successful Scapegoating, ensuring that if the system's incorrect final output is audited, the error is attributed to a benign downstream agent, the "scapegoat" $a_v$, rather than to the true malicious source, $a_m$.

With the system and attacker defined, we can now formulate the central problem that our SGoatMAS framework is designed to solve. The attacker's challenge is not simply to output malicious content, but to solve a complex optimization problem: where and how to inject a manipulative payload to maximize impact while minimizing personal risk. The "where" corresponds to selecting an optimal target link in the workflow, and the "how" corresponds to crafting the payload. We formally define the attacker's problem as finding an optimal attack plan $\pi^*$. An attack plan $\pi$ consists of a target link $e = (a_u, a_v)$ and a payload $c_u$ that agent $a_u$ is manipulated to produce. The optimal plan $\pi^* = (e^*, c_u^*)$ is one that solves the following multi-objective optimization problem:

$$\pi^* = \arg\max_{\pi} \left[ \mathbb{P}(\mathcal{O}_m | \pi) - \lambda \cdot \mathbb{P}(\text{Attribution}(a_m) | \pi) \right] \tag{2}$$

In this equation, $\mathbb{P}(\mathcal{O}_m | \pi)$ is the probability of the malicious objective being achieved given the execution of attack plan $\pi$, and $\mathbb{P}(\text{Attribution}(a_m) | \pi)$ is the probability of the attack being correctly attributed to the malicious agent $a_m$. The parameter $\lambda \in [0, \infty)$ represents the attacker's risk aver-sion; a higher $\lambda$ indicates a greater preference for stealth over success.

Directly solving the optimization problem posed in Equation 2 is intractable, as the probabilities depend on the complex, opaque internal states of the LLM agents and the intricate dynamics of their interaction. Therefore, our SGoatMAS framework, detailed in the following section, is designed to provide a tractable and effective heuristic for approximating this optimal attack plan. It achieves this by first identifying the optimal target link $e^*$ by maximizing a carefully designed proxy metric—the Vulnerability-to-Risk Ratio —which is engineered to correlate strongly with the objective function in Equation 2, thereby balancing the competing demands of attack efficacy and covertness.

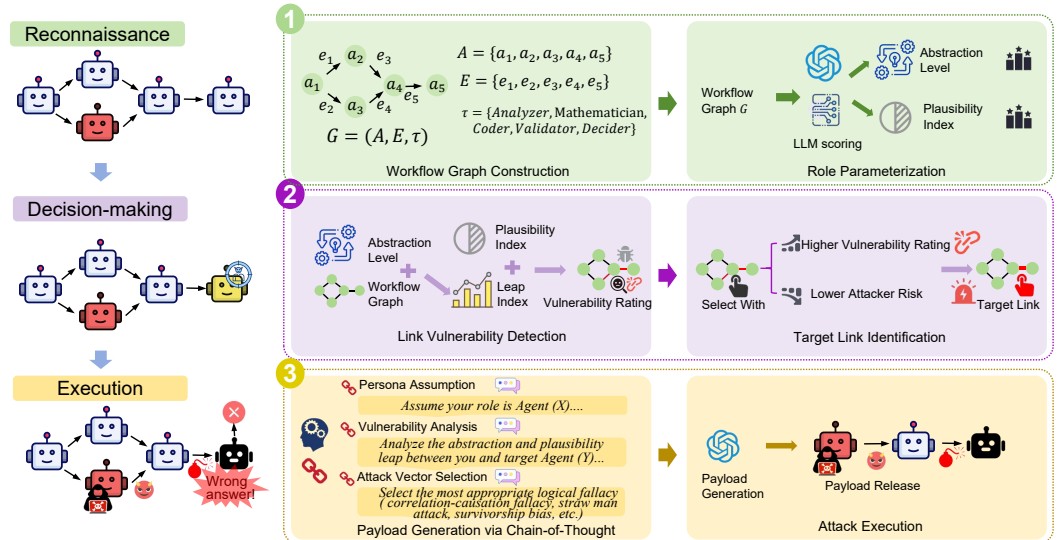

Figure 1: Workflow of SGoatMAS. The attack proceeds in three stages: reconnaissance, decision-making, and execution. In reconnaissance, the attacker constructs a workflow graph from observations and uses an LLM to parameterize each role. During decision-making, vulnerability and risk indices are evaluated to identify the optimal attack target link via the VRR metric. In execution, the attacker generates a suitable payload via a Chain-of-Thought template with an LLM and launches the attack.

## 4  THE SGOATMAS FRAMEWORK

We propose SGoatMAS, a formal framework for covert Scapegoating attacks within LLM-based Multi-Agent Systems. The framework guides a malicious agent, $a_m$, through a principled process to identify the most strategic point of attack and to craft a subtle, logic-based payload to induce an error in a designated scapegoat. As illustrated in Figure 1, the attack proceeds in three stages: (1) Reconnaissance, where the system is modeled and vulnerabilities are parameterized; (2) Decision-Making, where the optimal attack vector is mathematically identified; and (3) Execution, where the malicious payload is generated and deployed. The process is detailed in Algorithm 1.

### 4.1  STAGE 1: RECONNAISSANCE AND VULNERABILITY PARAMETERIZATION

The first stage involves passive observation and modeling of the target MAS. The attacker aims to create a "cognitive map" of the system, quantifying potential vulnerabilities of each agent's role.

#### 4.1.1  WORKFLOW GRAPH CONSTRUCTION

The attacker first observes the agent interactions to model the system as a DAG $G = (A, E, \tau)$, as defined in Section 3. This establishes the topological foundation for the attack, identifying the agents $A$ and the information flow pathways $E$.

#### 4.1.2  CONTEXT-AWARE ROLE PARAMETERIZATION

A critical challenge in attacking a semantic system is to move beyond topology and quantify the cognitive properties of each agent's role. To mitigate attacker bias and ensure a reproducible strategy, we propose a context-aware evaluation method using a powerful, independent LLM (e.g., GPT-4) as an expert assessor. This evaluator is given the full description of the system's goal and all agent roles. It is then tasked with assigning two key scores to each role $\tau$:

Abstraction Level ($\text{Abs}(\tau)$): This score quantifies how far a role's primary output is from raw data. Its theoretical basis is the DIKW hierarchy (Data → Information → Knowledge → Wisdom) (Ackoff, 1989), a well-established model of information processing. A role that transforms raw data into

---

**Algorithm 1** SGoatMAS Attack Planning

---

**Require:** Workflow Graph $G = (A, E)$, Malicious Agent $a_m$
1: **// Stage 1: Reconnaissance**
2: Let $\mathcal{P} \leftarrow \emptyset$           ▷ Initialize parameter map
3: **for** each agent $a_i \in A$ **do**
4:    $\mathcal{P}[a_i] \leftarrow \text{LLMEvaluator}(\tau(a_i), G, \{\tau(a_j)\}_{j=1..n})$
5: **end for**
6:
7: **// Stage 2: Decision-Making**
8: Initialize max_vrr $\leftarrow -\infty$, $e^* \leftarrow$ null
9: **for** each edge $e = (a_u, a_v) \in E$ **do**
10:    $L \leftarrow 1 + \max(0, \mathcal{P}[a_v].\text{Abs} - \mathcal{P}[a_u].\text{Abs})$
11:    $P \leftarrow \mathcal{P}[a_v].\text{Amb}$
12:    $V(e) \leftarrow L \cdot P$
13:    $R(e|a_m) \leftarrow 1/(d(a_m, a_u) + 1)$
14:    vrr $\leftarrow V(e)/(R(e|a_m) + 1e - 9)$
15:    **if** vrr $>$ max_vrr **then**
16:      max_vrr $\leftarrow$ vrr
17:      $e^* \leftarrow e$
18:    **end if**
19: **end for**
20:
21: **// Stage 3: Execution**
22: prompt $\leftarrow \text{GenerateCoTPrompt}(e^*, \mathcal{O}_m)$
23: **return** $e^*$, prompt        ▷ Return target and payload prompt

---

a structured table has a low abstraction level, while a role that synthesizes multiple reports into a strategic recommendation has a high one.

Task Ambiguity ($\text{Amb}(\tau)$): This score measures the degree to which a role's task lacks a single, objectively verifiable correct answer. It is inspired by Heider's Attribution Theory (Heider, 2013), which suggests that blame for a failure is more plausibly attributed to roles with high intrinsic uncertainty. A code-checking agent has low ambiguity, while a brainstorming agent has high ambiguity.

The output of this stage is a fully parameterized graph where each node (agent) is annotated with its cognitive properties, preparing the ground for strategic decision-making.

### 4.2 STAGE 2: DECISION-MAKING VIA VULNERABILITY-TO-RISK RATIO (VRR)

With the system mapped and its components parameterized, the attacker must now decide where to strike. The cornerstone of SGoatMAS is a quantitative evaluation that balances the potential impact of an attack against the risk of detection. To achieve this, we introduce a principled metric, the Vulnerability-to-Risk Ratio, to identify the most strategically advantageous point of attack. This involves evaluating the vulnerability of each link and the associated risk to the attacker.

#### 4.2.1 LINK VULNERABILITY, $V(e)$

First, we define the inherent vulnerability of a link, $V(e)$, as a function of both the cognitive challenge it presents and the plausibility of an error occurring. We model this as a multiplicative interaction, implying that a link is only highly vulnerable if both conditions are met:

$$V(e) = \text{Leap}(e) \cdot \text{Plausibility}(e) \tag{3}$$

The Cognitive Leap Index, $\text{Leap}(e)$, quantifies the magnitude of the abstractive transformation across the link, drawing from Sweller's Cognitive Load Theory (Sweller, 1988). Large cognitive leaps are more mentally taxing and thus more prone to manipulation.

$$\text{Leap}(e) = 1 + \max(0, \text{Abs}(\tau(a_v)) - \text{Abs}(\tau(a_u))) \tag{4}$$

The Error Plausibility Index, $\text{Plausibility}(e)$, measures the perceived reasonableness of an error occurring at the downstream agent $a_v$, which is crucial for successful scapegoating.

$$\text{Plausibility}(e) = \text{Amb}(\tau(a_v)) \tag{5}$$

### 4.2.2 ATTACKER RISK, $R(e|a_m)$

Second, we define the risk to the attacker, $R(e|a_m)$, as a function of being detected and attributed. In our simplified model, we equate this risk with the traceability of the information path, based on the forensic principle that traceability decays with distance.

$$R(e|a_m) = \text{Traceability}(e|a_m) = \frac{1}{d(a_m, a_u) + 1} \quad (6)$$

where $d(a_m, a_u)$ is the length of the shortest path in the graph from the malicious agent $a_m$ to the link's upstream agent $a_u$. A shorter path implies a higher risk of attribution.

### 4.2.3 OPTIMAL TARGET SELECTION WITH VRR

Having defined the vulnerability and risk components, we can now formulate the final decision metric. The Vulnerability-to-Risk Ratio for any given link $e = (a_u, a_v)$ is defined as:

$$\text{VRR}(e|a_m) = \frac{V(e)}{R(e|a_m) + \epsilon} \quad (7)$$

where $V(e)$ and $R(e|a_m)$ are calculated as above, and $\epsilon$ is a small constant to prevent division by zero. This ratio formulation captures the attacker's strategic goal: to maximize impact (vulnerability) for a given level of risk. The SGoatMAS framework identifies the optimal target link, $e^*$, by selecting the link that maximizes this score:

$$e^* = \arg\max_{e \in E} \text{VRR}(e|a_m) \quad (8)$$

This link represents the most strategic point in the workflow to inject a malicious payload.

### 4.3 STAGE 3: EXECUTION VIA COT-BASED PAYLOAD GENERATION

Once the optimal target link $e^* = (a_u, a_v)$ is identified, the final stage is to craft and deploy the malicious payload. The goal is to generate a payload that is logically flawed but semantically plausible. To achieve this, we leverage the reasoning capabilities of Chain-of-Thought prompting (Wei et al., 2022). While typically used to elicit correct reasoning (Kojima et al., 2022), we repurpose CoT as an offensive tool to construct a compelling, yet fallacious, justification for a flawed piece of information. The SGoatMAS framework generates a specialized CoT prompt that instructs a powerful LLM to perform a structured reasoning sequence. The core steps of this guided reasoning process are outlined below:

---

**CoT Prompt Logic for Payload Generation**

1. **Assume Persona:** Adopt the role of the upstream agent, $a_u$.

2. **Analyze Vulnerability:** Reason about the specific cognitive leap between $\tau(a_u)$ and $\tau(a_v)$.

3. **Select Logical Fallacy:** Choose a subtle fallacy (e.g., base rate neglect, correlation-causation, straw man attack, survivorship bias, etc.) that best exploits this leap.

4. **Craft Payload:** Generate a final, professionally-worded output based on the fallacious reasoning path.

---

This process produces a "logical bomb"—information not easily disproven by fact-checking but designed to detonate within the reasoning of the scapegoat agent, $a_v$, thereby executing the attack.

## 5 EXPERIMENTS

We conduct comprehensive experiments to validate the efficacy, stealth, and strategic capabilities of the SGoatMAS framework. Our evaluation is designed to answer four key research questions:

- **RQ1 (Effectiveness):** How effectively can SGoatMAS degrade the performance of a MAS on complex reasoning tasks compared to a direct, naive attack?

- **RQ2 (Stealth & Scapegoating):** What is the trade-off between SGoatMAS's effectiveness, its detection rate, and its ability to successfully frame a scapegoat?

- **RQ3 (Generalizability):** How does the attack's performance vary across different MAS workflow topologies?

- **RQ4 (Strategic Importance):** How does the malicious agent's specific position within the workflow critically impact the attack's outcome, and how do these impacts differ across topologies?

## 5.1 EXPERIMENTAL SETUP

### 5.1.1 DATASETS AND TASKS

We evaluate our framework on a diverse set of five widely-used benchmarks to test a range of capabilities:

- General knowledge & reasoning: MMLU (Hendrycks et al., 2020) and its more challenging successor, MMLU-Pro (Wang et al., 2024d).

- Mathematical reasoning: GSM8K (Cobbe et al., 2021) and Arithmetic (Du et al., 2023).

- Code generation: HumanEval (Chen et al., 2021).

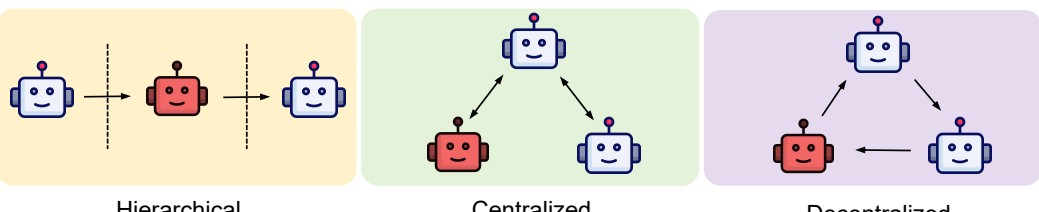

Hierarchical        Centralized        Decentralized

Figure 2: The three multi-agent workflow topologies used in our experiments: (a) Hierarchical, (b) Centralized (workers only communicate to the central manager directly), and (c) Decentralized.

### 5.1.2 MULTI-AGENT WORKFLOW TOPOLOGIES

We implement three distinct, three-agent workflow topologies $(A_1, A_2, A_3)$: Hierarchical, Centralized, and Decentralized, as illustrated in Figure 2.

### 5.1.3 ATTACK SCENARIOS & BASELINES

We evaluate three scenarios: a Benign baseline (no attack), a Naive Attack baseline (direct, forceful attack), and our proposed SGoatMAS Attack. In the attack scenarios, one agent ($A_1$, $A_2$, or $A_3$) is designated as the malicious agent, $a_m$. All agents in this study employ the GPT-4o-mini API.

### 5.1.4 EVALUATION METRICS

We define three key metrics: Accuracy (Acc.), the attacker's Detection Rate (DR), and the Scapegoat Accuracy (SA). The complete dialogue of all agents within each workflow is recorded, and upon workflow completion, these records are provided to a detect agent for identifying the most suspicious agent based on the conversation history. For evaluation, 50 samples are randomly selected from each dataset. In each attack experiment, if the detect agent correctly identifies the malicious agent, the result is recorded as Detection Rate; if it fails and instead misclassifies a benign agent, the result is recorded as Scapegoat Accuracy.

## 5.2 MAIN RESULTS: COMPARATIVE ANALYSIS OF ATTACK METHODOLOGIES

To answer RQ1 and RQ2, we first present a comparison of attack methodologies in Table 1. The table presents a direct comparison of the Benign, Naive, and SGoatMAS scenarios for each dataset,

Table 1: Comprehensive performance evaluation across all datasets, topologies, and attack methodologies. Abbreviations: H. (Hierarchical), C. (Centralized), D. (Decentralized). Each cell reports three key metrics: final system Accuracy (Acc. %), Detection Rate (DR %), and Scapegoat Accuracy (SA %). For each metric, the best-performing attack method (lowest Acc., lowest DR, highest SA) is highlighted in bold.

| Topo. | Method | MMLU | | | MMLU-Pro | | | GSM8K | | | Arithmetic | | | HumanEval | | |
|---|---|---|---|---|---|---|---|---|---|---|---|---|---|---|---|---|
| | | Acc. | DR | SA | Acc. | DR | SA | Acc. | DR | SA | Acc. | DR | SA | Acc. | DR | SA |
| H. | Benign | 90.0 | – | – | 62.0 | – | – | 67.3 | – | – | 23.3 | – | – | 76.7 | – | – |
| | Naive Attack | 54.7 | 78.0 | 22.0 | 30.7 | 73.3 | 26.7 | 36.0 | 79.3 | 20.7 | **8.0** | 72.7 | 27.3 | 52.0 | 85.3 | 14.7 |
| | SGoatMAS | **46.0** | **66.0** | **34.0** | **20.0** | **60.0** | **40.0** | **34.0** | **58.7** | **41.3** | **8.0** | **52.7** | **47.3** | **50.7** | **58.0** | **42.0** |
| C. | Benign | 82.0 | – | – | 51.3 | – | – | 62.0 | – | – | 20.0 | – | – | 58.0 | – | – |
| | Naive Attack | 52.0 | 83.3 | 16.7 | 32.7 | 83.3 | 16.7 | 39.3 | 87.3 | 12.7 | 6.7 | 76.7 | 23.3 | **41.3** | 96.0 | 4.0 |
| | SGoatMAS | **48.7** | **48.7** | **51.3** | **31.3** | **43.3** | **56.7** | **34.7** | **61.3** | **38.7** | **4.7** | **61.3** | **38.7** | **41.3** | **72.0** | **28.0** |
| D. | Benign | 82.0 | – | – | 52.7 | – | – | 62.0 | – | – | 19.3 | – | – | 82.7 | – | – |
| | Naive Attack | 72.7 | 94.0 | 6.0 | **46.7** | 91.3 | 8.7 | 30.7 | 86.7 | 13.3 | 8.7 | 84.0 | 16.0 | 81.3 | 73.3 | 26.7 |
| | SGoatMAS | **65.3** | **83.3** | **16.7** | 47.3 | **84.7** | **15.3** | **20.7** | **83.3** | **16.7** | **8.0** | **82.0** | **18.0** | **74.0** | **60.7** | **39.3** |

with results grouped by workflow topology. To provide a general overview, the data for the attack scenarios is averaged over all three potential malicious agent positions.

We find that SGoatMAS effectively reduces system accuracy—by up to 40%—while achieving lower detection rates and higher Scapegoat Accuracy than a Naive Attack (detection can drop by up to 40% relative to the naive baseline). Thus, for a given level of performance degradation, SGoat-MAS substantially increases stealth, enabling more persistent and damaging attacks.

Attack effectiveness varies with workflow topology. Hierarchical and Centralized structures produce stronger effects than Decentralized ones, likely because final/central agents determine the outcome: convincing a single final/central agent suffices to sway the result (and if the malicious agent occupies that role, its impact is maximal). In Decentralized workflows the outcome is determined by majority after discussion, so an attacker must push a much stronger payload to shift consensus, which is harder.

Across datasets, achieving both a large accuracy drop and stealth is harder on knowledge-based QA sets (e.g., MMLU, MMLU-Pro) than on mathematical reasoning sets (e.g., GSM8K, Arithmetic). This likely reflects that knowledge QA items are often answerable independently (requiring less multi-step collaboration), making injected misinformation easier to detect or correct.

### 5.3 ABLATION STUDY: IMPACT OF POSITION, TOPOLOGY, AND SCALE

We now conduct a series of deep-dive ablation studies to answer RQ3 and RQ4, exploring the impact of the attacker's position, the system's topology, and the scale of the agent population.

#### 5.3.1 IMPACT OF ATTACKER'S POSITION AND TOPOLOGY

Table 2 presents the full, disaggregated results of the SGoatMAS attack, showing its performance for every combination of malicious agent position ($a_m$) and workflow topology across all five datasets.

We observe markedly different attack outcomes depending on the malicious agent's position in Hierarchical and Centralized topologies. If the attacker occupies the final position in a Hierarchical workflow or the central manager in a Centralized topology, system accuracy can collapse to near zero (up to a 90% drop) while maintaining strong stealth. Hence placing untrusted agents in these roles poses a substantial security risk for MAS. By contrast, when the attacker is in the first position of a Hierarchy or is an ordinary worker in a Decentralized topology—especially when the hierarchical path is long or the easy-to-answer dataset (e.g., MMLU) allows agents to judge answers independently—the attack effect is much weaker. In Decentralized settings, where roles are more egalitarian, the variance in attack effectiveness across positions is correspondingly small.

Table 2: Ablation study of SGoatMAS attack performance, broken down by malicious agent position ($a_m$) and workflow topology. Acc. (%): Final Accuracy of the SGoatMAS attack. DR (%): Detection Rate of the SGoatMAS attack. SA (%): Scapegoat Accuracy.

| Dataset | Metric | Hierarchical | | | Centralized | | | Decentralized | | |
|---|---|---|---|---|---|---|---|---|---|---|
| | | $A_1$ | $A_2$ | $A_3$ | $A_1$ | $A_2$ | $A_3$(central) | $A_1$ | $A_2$ | $A_3$ |
| **MMLU** | Acc. (%) ($\Delta P$) | 78.0 (-12.0) | 60.0 (-30.0) | 0.0 (-90.0) | 72.0 (-10.0) | 70.0 (-12.0) | 4.0 (-78.0) | 62.0 (-22.0) | 58.0 (-22.0) | 76.0 (-6.0) |
| | DR (%) ($\Delta$) | 44.0 (-0) | 8.0 (-10.0) | 74.0 (-26.0) | 44.0 (-28.0) | 48.0 (-32.0) | 54.0 (-44.0) | 82.0 (-8.0) | 80.0 (-14.0) | 88.0 (-10.0) |
| | SA (%) | 56.0 | 92.0 | 26.0 | 56.0 | 52.0 | 46.0 | 18.0 | 20.0 | 12.0 |
| **MMLU-Pro** | Acc. (%) ($\Delta P$) | 40.0 (-20.0) | 20.0 (-40.0) | 0.0 (-66.0) | 44.0 (-6.0) | 46.0 (+6.0) | 4.0 (-48.0) | 48.0 (-6.0) | 48.0 (-4.0) | 46.0 (-6.0) |
| | DR (%) ($\Delta$) | 30.0 (-0) | 70.0 (-20.0) | 80.0 (-20.0) | 48.0 (-32.0) | 52.0 (-22.0) | 30.0 (-66.0) | 84.0 (-8.0) | 76.0 (-10.0) | 94.0 (-2.0) |
| | SA (%) | 70.0 | 30.0 | 20.0 | 52.0 | 48.0 | 70.0 | 16.0 | 24.0 | 6.0 |
| **GSM8K** | Acc. (%) ($\Delta P$) | 54.0 (-14.0) | 48.0 (-20.0) | 0.0 (-66.0) | 50.0 (-10.0) | 52.0 (-12.0) | 2.0 (-60.0) | 32.0 (-34.0) | 8.0 (-52.0) | 22.0 (-38.0) |
| | DR (%) ($\Delta$) | 20.0 (-32.0) | 70.0 (-16.0) | 86.0 (-14.0) | 60.0 (-22.0) | 62.0 (-18.0) | 62.0 (-38.0) | 76.0 (-8.0) | 88.0 (-2.0) | 86.0 (-0) |
| | SA (%) | 80.0 | 30.0 | 14.0 | 40.0 | 38.0 | 38.0 | 24.0 | 12.0 | 14.0 |
| **Arithmetic** | Acc. (%) ($\Delta P$) | 14.0 (-10.0) | 10.0 (-14.0) | 0.0 (-22.0) | 8.0 (-12.0) | 6.0 (-14.0) | 0.0 (-20.0) | 8.0 (-12.0) | 10.0 (-10.0) | 6.0 (-12.0) |
| | DR (%) ($\Delta$) | 6.0 (-36.0) | 70.0 (-12.0) | 82.0 (-12.0) | 68.0 (-12.0) | 70.0 (-10.0) | 46.0 (-24.0) | 80.0 (-2.0) | 82.0 (-4.0) | 84.0 (-0) |
| | SA (%) | 94.0 | 30.0 | 18.0 | 32.0 | 30.0 | 54.0 | 20.0 | 18.0 | 16.0 |
| **HumanEval** | Acc. (%) ($\Delta P$) | 70.0 (-8.0) | 78.0 (+2.0) | 4.0 (-72.0) | 60.0 (+2.0) | 54.0 (-2.0) | 10.0 (-50.0) | 78.0 (-6.0) | 74.0 (-12.0) | 70.0 (-8.0) |
| | DR (%) ($\Delta$) | 2.0 (-54.0) | 82.0 (-18.0) | 90.0 (-10.0) | 70.0 (-18.0) | 90.0 (-10.0) | 56.0 (-44.0) | 92.0 (-8.0) | 90.0 (-10.0) | 0.0 (-20.0) |
| | SA (%) | 98.0 | 18.0 | 10.0 | 30.0 | 10.0 | 44.0 | 8.0 | 10.0 | 100.0 |

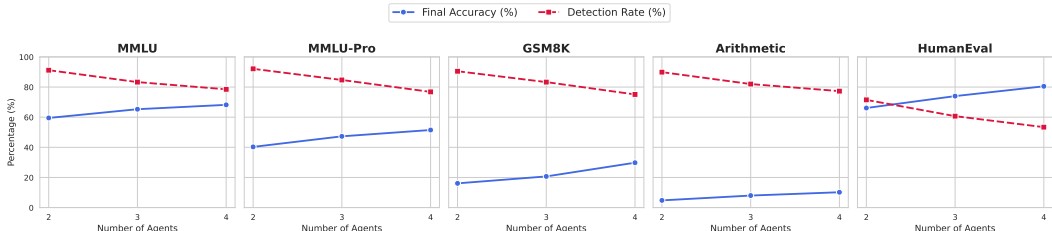

Figure 3: Impact of system scale in decentralized topologies on SGoatMAS Attack Outcomes.

### 5.3.2 IMPACT OF SYSTEM SCALE

Finally, to explore the relationship between system size and attack stealth, we conducted an additional experiment on the decentralized topology, varying the number of participating agents from two to four. As shown in Figure 3, a clear trade-off emerges. As the number of agents increases, the final system accuracy under attack gradually rises (i.e., the attack's direct impact is diluted across more agents). However, the attacker's Detection Rate drops significantly. This suggests that larger, more populated agent systems, while potentially more robust to direct sabotage, offer greater anonymity for a covert attacker. This "hiding in the crowd" effect makes inducement attacks an even more potent and insidious threat in large-scale systems.

## 6 CONCLUSION

In this paper, we investigated a novel, covert attack strategy in Multi-Agent Systems, termed the *Scapegoating attack*. Our goal was to devise a method by which a malicious internal agent could sabotage a task's objective while avoiding attribution. To this end, we proposed the SGoatMAS framework. The methodology is centered on two primary contributions: first, the conceptualization of the MAS workflow as a System-level Chain of Thought, which allows for the identification of systemic vulnerabilities between agents; and second, the introduction of the Vulnerability-to-Risk Ratio (VRR), a quantitative metric to identify the optimal attack vector by balancing system vulnerability against attacker risk. Our extensive experiments confirm that the SGoatMAS framework is effective at significantly degrading system performance while maintaining low detectability. This work provides the first formal methodology for executing inducement-based attacks and highlights the need for system-aware, rather than agent-centric, security defenses in MAS.

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
