# OpenReview forum: "You Don't Have to Be the One Doing Evil! Locate a Scapegoat within Multi-Agent Systems for Executing Covert Attacks"
_ICLR.cc/2026/Conference — ICLR 2026 Conference Withdrawn Submission_

### Official Review · Reviewer_3W26 · 2025-10-26

**Soundness:** 3
**Presentation:** 2
**Contribution:** 2
**Rating:** 2
**Confidence:** 4

**Summary:**

This paper introduced the Scapegoating attack, wherein malicious agents do not directly perpetrate harm but instead induce downstream agents to output harmful/erroneous content, thereby shifting blame onto a scapegoat during audits. The authors consequently proposed the SGoatMAS framework, treating multi-agent workflows as a System-level CoT, selecting the most vulnerable yet covert attack chain using VRR, and generating ‘plausible-sounding’ logical payloads via CoT prompts. Experiments across five benchmarks and three topologies demonstrate that while reducing system accuracy, the approach significantly lowers detection rates and increases the probability of scapegoating. However, insufficient proposal details, limited experimental analysis, and ambiguous attack procedures constrain the contributions of this paper.

**Strengths:**

1. The threat model is novel: shifting from ‘direct attacks’ to ‘induction + blame-shifting’ increases audit complexity; treating the MAS inference chain as a system-level CoT offers an insightful perspective.

2. Stealthy: Compared to the Naive attack, detection rates are significantly reduced, scapegoat hit rates are increased, while system accuracy shows a marked decline.

**Weaknesses:**

1. Insufficient verifiability of VRR: Key parameters (role “abstraction/ambiguity”) are scored by external LLMs, introducing potential subjectivity and model dependency; lacking ablation studies on evaluator sensitivity.

2. Detection and scapegoat assessment may be biased: detectors are all LLM-based (single template/single model), with no human audit controls or multi-detector integration observed; SA metrics may overestimate actual scapegoating effects.

3. Evaluation limitation: All agents uniformly employ a closed-source API (GPT-4o-mini), with each dataset comprising only 50 samples; cross-validation across different foundational models or system implementations is absent.

4. Discussion of defensive aspects is insufficient: whilst new attack surfaces are identified, systematic experimentation on mitigation/defence measures is lacking.

5. There is a lack of threat modelling, as well as a lack of the attacker's capabilities and knowledge.

6. Unpublished system prompts/role prompts/attack CoT templates and parameters (temperature, top-p, max tokens, seed, retry strategy) make it difficult to reproduce/verify conclusions regarding the trade-off between ‘stealth and effectiveness’. Key metrics (Abs/Amb scoring, constants in the VRR formula, and risk aversion coefficients ϵ and λ) lack more experiment analysis and discussion.

7. There is no case-by-case analysis reconstructing the specific language/causal chain of ‘how to shift blame downstream’; the ordering stability/interpretability of VRR has not been systematically verified. The claim that it generates disruptive content rather than disruptive behaviour requires further clarification.

8. Scale and Network Structure: Extending the number of decentralised agents to 5–9 to verify whether the “group stealth effect” continues to intensify; the impact of multi-community/small-world structures on propagation.

9. This paper lacks supporting material, particularly usable code, resulting in reduced reproducibility.

**Questions:**

1. Cross-model/cross-framework: Can the results be replicated across open-source LLMs (such as the Llama series) and different MAS frameworks?

2. Is a sample size of 50 per dataset sufficient to support conclusions?

Suggestions

1. format error: ``System-level Chain of Thought'' instead of ”System-level Chain of Thought”

2. Metrics should employ arrows to denote the direction of advantage.

---

### Official Review · Reviewer_FQsy · 2025-10-28

**Soundness:** 3
**Presentation:** 3
**Contribution:** 3
**Rating:** 6
**Confidence:** 4

**Summary:**

This paper introduces a novel covert attack strategy for LLM-based Multi-Agent Systems called the "Scapegoating attack"， which has a malicious agent inducing a downstream, benign agent (the "scapegoat") to produce the harmful output, allowing the true attacker to remain undetected .The authors present SGoatMAS, the systematic framework that models MAS workflows as "System-level Chain of Thought", introduces the Vulnerability-to-Risk Ratio (VRR) metric to identify optimal attack points that maximize impact while minimizing the attacker's risk, and leverages Chain-of-Thought prompting to generate subtle, logic-based payloads. Experiments on five benchmarks across three different MAS workflow topologies demonstrate significant accuracy degradation (up to 40%) with lower detection rates (up to 40%) compared to naive attacks.

**Strengths:**

1. High Originality and Significance: This paper introduces a new class of attacks, the "Scapegoating attack" , which is more subtle and dangerous than the direct attacks commonly studied. This work effectively defines a systematic red-teaming framework and evaluation setting for MAS security.

2. Rigorous Experimental Design: The paper's experimental validation tests the SGoatMAS framework across five diverse task domains (MMLU, MMLU-Pro, GSM8K, Arithmetic, HumanEval) and three distinct Multi-Agent System topologies (Hierarchical, Centralized, Decentralized). This cross-domain and topology-based design powerfully demonstrates the attack's generalizability and practical utility.

3. Multi-Factor Ablation Studies: The paper includes crucial ablation studies analyzing the impact of the attacker's position within the workflow (Table 2) and the effect of system scale (Figure 3), which reveals the non-obvious "hiding in the crowd" phenomenon.

**Weaknesses:**

1. Dependency on a Powerful LLM Evaluator: The paper relies heavily on a "powerful, independent LLM (e.g., GPT-4)" both as the Stage-1 assessor for scoring $Abs(\tau)$/$Amb(\tau)$ and as the CoT-based payload generator in Stage-3. This framework's effectiveness seems dependent on access to a highly capable model for the evaluation and the generation. It would be useful to know how sensitive the SGoatMAS framework is to the quality of this evaluator. Does the attack still work if the evaluator is a less-capable model? Moreover, the experiments use GPT-4o-mini for all task agents while recommending a stronger model (GPT-4) as the assessor, which may overstate attacker realism when defenders run comparable or stronger models.

2. Overreliance on Simplified MAS Topologies: Most experiments use fixed 3-agent topologies; although a small-scale extension (2–4 agents) is included for decentralized settings, evaluation on larger, heterogeneous, and dynamic MAS would improve external validity.

3. Small per-dataset sample size: The evaluation uses only 50 randomly selected samples per dataset, which risks high sampling variance. To increase statistical reliability, the authors should report confidence intervals or repeat experiments with multiple random seeds (or larger sample sizes) so readers can gauge result stability.

4. Lack of Realistic Defense or Countermeasure Discussion: While the paper proposes a sophisticated attack, it provides no discussion or evaluation of possible defenses. Demonstrating at least preliminary mitigation strategies (e.g., anomaly detection, role verification, or consistency auditing) would strengthen the contribution and ethical grounding.

5. Evaluation Metrics May Be Circular: The "Detection Rate" and "Scapegoat Accuracy" rely on a detect agent that is itself an LLM classifier. This raises concerns about bias and consistency—different LLMs or prompt styles could yield drastically different detection outcomes. And the paper does not specify the exact model/prompting used for the detect agent, which may make DR/SA sensitive to detector choice and hinder reproducibility.

**Questions:**

See Weaknesses.

---

### Official Review · Reviewer_652U · 2025-10-29

**Soundness:** 3
**Presentation:** 3
**Contribution:** 3
**Rating:** 4
**Confidence:** 4

**Summary:**

This paper presents a attack 'the Scapegoating attack' in MAS. The goal is for a malicious agent to secretly cause a task failure by making another normal agent 'scapegoat' produce the wrong output. Authors design SGoatMAS, a framework that treats the MAS workflow as a DAG. It uses external LLM to set agent roles. SGoatMAS finds the best attack link by maximizing the VRR. The framework then uses CoT  with another LLM to create message that misleads scapegoat. Experiments show that SGoatMAS lowers task accuracy, reduces detection rate, and raises scapegoat identification rate.

**Strengths:**

- Defines the 'Scapegoating' attack. It shows a new risk: secret manipulation and blame-shift in MAS beyond just direct sabotage.

- SGoatMAS is a clear multi-stage system.

- Shows a new use of CoT. CoT normally helps reasoning. Here it makes flawed yet believable attack payloads.

- Experiments cover different reasoning types.

**Weaknesses:**

- The VRR components are based on weak theory and lacks solid evaluation.

- SGoatMAS uses LLMs for scoring and CoT. This makes it costly and hard to repeat. It depends on the LLMs' ability and biases. The stability of the role scoring is not evaluated.

- Experiments use three agents only. Real MAS has many more agents.

- The paper does not introduce how it picks which logical fallacy. It also doesnot show how to tune the CoT prompt. These steps are not described well and hard to follow.

- The detection mechanism (a single agent reading logs) seems weak.

**Questions:**

1. See weakness

2. Can you provide stronger justification for the specific formulations of $Leap(e)$, $Plausibility(e)$, and $R(e|a_m)$?

3. How robust is the LLM-based role parameterization (Sec 4.1.2)?

---

### Official Review · Reviewer_iqBR · 2025-10-29

**Soundness:** 2
**Presentation:** 3
**Contribution:** 2
**Rating:** 2
**Confidence:** 3

**Summary:**

The authors introduce the Scapegoating attack in which a malicious agent in a MAS compels downstream agents to sabotage task completion, thereby 'deflecting blame.' They implement the attack with the SGoatMAS framework, a three step MAS poisoning process. Evaluations are performed on three datasets, three topologies, using the GPT-4o-mini model

**Strengths:**

Well written paper that addresses a clear and interesting problem.

**Weaknesses:**

- **Threat Model:** The attack requires a malicious agent to be (1) within a system, (2) observe rollouts that are possibly query-dependent, and (3) observe system prompts / personas of other agents. Similar attacks (like Control Flow Hijacks from Triedman et al.) similarly deflect blame with far fewer assumptions (only access to external information). I would like to see some justification to the threat model and why this model is valuably distinct from the traditional IPI setup.

- **Evaluation Metric:** The core metric evaluated is detection rate. The detector, to my understanding, is a prompted gpt-4o-mini model. This model is quite old (as far as frontier LMs go) and doesn't handle complex context well. Given that the presented detection rates are non-negligibly high with this model, I would like to see evaluations with a better model (e.g., gpt-4o, gpt-5, o4, SecAlign) and a more established detection method (e.g., LlamaFirewall).

- **Agent Model:** Would agents armed with a better model be susceptible to this attack?

- **Baseline:** The Naive baseline is "direct, forceful" and little other details are provided. There are a lot of techniques from the indirect prompt injection literature that could apply here (e.g., control flow hijacking from Triedman et al.) to make a baseline more "covert." Triedman et al., also provides a blueprint to deflect blame. Stronger baselines, like this one, should be evaluated.

- **Method Complexity:** The method consists of three steps including rollout observation and the introduction of a new metric. It's not yet clear to me that either of these steps is necessary, especially since the existing naive baselines perform relatively well. I would like to see an ablation here.

**Questions:**

- Would agents armed with a better model be susceptible to this attack?
- Would a more sound detector detect your attack more often?
- What new threats does this threat model introduce that the traditional IPI model does not?

---

### Note · Authors · 2025-12-03

I have read and agree with the venue's withdrawal policy on behalf of myself and my co-authors.